Microbiota associated with urban forests

Wan Xin 1
Zhou Runyang 2
Yuan Yingdan 2
Xing Wei 695362718@qq.com 1
Liu Sian sianliu@yzu.edu.cn 2
1 Jiangsu Academy of Forestry , Nanjing , China
2 Yangzhou University , Yangzhou , China
Kumar Ravinder
Electronic publication date: 2024 Feb 29
Publication date: 2024
Volume: 12
Electronic Location ID: e16987
Received 2023 Oct 30; Accepted 2024 Jan 30
Copyright: ©2024 Wan et al.
Copyright year: 2024
Copyright holder: Wan et al.
License: This is an open access article distributed under the terms of the Creative Commons Attribution License, which permits unrestricted use, distribution, reproduction and adaptation in any medium and for any purpose provided that it is properly attributed. For attribution, the original author(s), title, publication source (PeerJ) and either DOI or URL of the article must be cited.
License URL: https://creativecommons.org/licenses/by/4.0/

Keywords: Urban forest, Soil microbial, Tree species, Soil property, Community diversity

Funding: The Forestry Science and Technology Innovation and Promotion Project of Jiangsu Province ‘Long-term Research Base of Forest and Wetland Positioning Monitoring in Jiangsu Province’ No. LYKJ[2020]21 The Construction model of Efficient Farmland Protection Forest Network in Jiangsu Province No. LYKJ[2021]38 The Research on Efficiency Management Technology of Carbon Sequestration Forest in Jiangsu Coast No. LYKJ[2021]25 This study was funded by the Forestry Science and Technology Innovation and Promotion Project of Jiangsu Province ‘Long-term Research Base of Forest and Wetland Positioning Monitoring in Jiangsu Province’ (No. LYKJ[2020]21), the Construction model of Efficient Farmland Protection Forest Network in Jiangsu Province (No. LYKJ[2021]38) and the Research on Efficiency Management Technology of Carbon Sequestration Forest in Jiangsu Coast (No. LYKJ[2021]25). The funders had no role in study design, data collection and analysis, decision to publish, or preparation of the manuscript.

==============================
Urban forests are essential for maintaining urban ecological stability. As decomposers, soil microorganisms play an indispensable role in the stability of urban forest ecosystems, promoting the material cycle of the ecosystems. This study used high-throughput sequencing technology to explore the bacteria in six forest stands, including Phyllostachys edulis (ZL), Metasequoia glyptostroboides (SSL), Cornus officinalis (SZY), mixed broad-leaved shrub forest (ZKG), mixed pine and cypress forest (SBL), and mixed broad-leaved tree forest (ZKQ). Meanwhile, the differences in fungal communities were investigated. The results show that ZL has the highest alpha diversity of bacterial communities, while its fungal community is the lowest; Proteobacteria is the most abundant bacterial phylum in the six forest stands; ZKQ has the highest fungal diversity. In addition, soil microbial communities are affected by environmental factors. Soil pH, organic matter (SOM), and available phosphorus (AP) significantly influence the compositions of urban forest soil microbial communities. This study revealed the differences in bulk soil (BS) microbial community structures among six forest stands and the relationship between environmental factors and soil microbial communities, which has important guiding significance for creating healthy and stable urban forests with profound ecological benefits.

Introduction

Urban forest includes all woodlands, tree groups and their related infrastructure in cities and surrounding areas (Nowak, 1993). As a network system composed of ecological and social connections, they are vital for urban construction. The concept of urban forests has been proposed for decades. However, it still lacks a unified standard. In 1962, “Urban Forest” appeared for the first time in the American government’s outdoor recreation resource survey report (Seamans, 2013). Urban forests are an essential component of urban ecosystems. They regulate microclimate, adsorb particulate matter and harmful gases, provide ecological services such as recreation, and offer shelters for native species (Alvarez-Codoceo, Cerda & Perez-Quezada, 2021). Urban forests can purify the air, beautify the environment, regulate climate, conserve water sources, and maintain biodiversity.

Urban soil microorganisms are critical in maintaining the stability of urban ecosystems (Bond-Lamberty et al., 2018). As the fundamental carrier, urban forest soil provides forest trees with growth nutrients. The quality of soil directly affects the growth of forest trees and ecological landscapes (Sun et al., 2015). During urban development, changes in land use patterns prominently influence soil microorganisms. Research shows that soil microbial biomass carbon and nitrogen play a crucial role in nutrient transformation, ecosystem stability, and soil fertility. There is a significant difference in soil microbial biomass carbon and nitrogen content between urban and suburban forests, and the soil microbial biomass carbon in urban forests is remarkably higher than that in suburban forests (Chen et al., 2013; Mao et al., 2014). Soil microorganisms promote soil biogeochemical processes and affect soil carbon and nitrogen conversion rates (Xiong et al., 2014). Higher soil microbial community diversity has a stronger ability to resist environmental stress (Deng et al., 2016; Riah-Anglet et al., 2015). Proteobacteria, Acidobacteria, Bacteroidetes, and Actinobacteria are proven to be the dominant phyla of urban soil bacteria. The major phyla of soil fungi include Ascomycota, Basidiomycota, and Zygomycota (Tan et al., 2019). Although the compositions of urban soil microbial communities are highly similar at the phylum level, there are significant differences at other classification levels due to urbanization levels and changes in the urban environment (Tan et al., 2019).

Soil microbial diversity and species compositions vary with environmental changes. Regarding the soil microbial diversity of urban green space systems, soil pH is the primary influencing factor (Yan et al., 2016; Zhalnina et al., 2015); other factors such as season, climate, precipitation, and temperature can lead to the variation of soil bacterial communities in urban forests (Xu et al., 2014). The exploration of soil microorganisms and soil physical and chemical properties in urban green spaces verifies that soil microbial diversity is also affected by soil moisture content and soil sand content (Wang, Wu & Kumari, 2018). With the intensification of human activities, especially the impact of heavy metal ions on soil microorganisms, urban soil pollution has become severe. Soil fungi are more susceptible to heavy metal pollution than bacteria (Rex et al., 2018). In addition, soil organic matter pollution can tremendously affect soil microbial diversity (Peng et al., 2016). Urbanization is one of the key reasons for the loss of natural ecosystems and the damage to ecological environments (Grimm et al., 2008). Urbanization will abate soil fungal diversity and result in the homogeneity of soil fungal communities, while soil bacterial diversity is hardly affected by urbanization (Epp Schmidt et al., 2017). Li et al. (2018) believed that plant communities facilitated driving underground fungal communities, in turn, the impact of fungal-plant interactions helps to enhance grassland plant abundance (Bennett & Cahill Jr, 2016). Therefore, the relationship between environmental factors and soil microorganisms will change along with the living environment, and the effects of environmental factors on microorganisms vary with ecological environments.

Urban forests are essential for urban ecosystems. This study selected six forest stands in the Zhuyu Bay Scenic Area in Yangzhou City, Jiangsu Province, as research subjects: Phyllostachys edulis (ZL), Metasequoia glyptostroboides (SSL), Cornus officinalis (SZY), mixed broad-leaved shrub forest (ZKG), mixed pine and cypress forest (SBL), and mixed broad-leaved tree forest (ZKQ) (Table 1), mainly focusing on two questions: (1) What commonalities and variances lie in the compositions of microbial communities in the bulk soil (BS) of six distinct forest stands? (2) How are soil factors related to the compositions of microbial communities in the BS? This study investigated soil microbial communities in six urban forest locations to understand the interaction between soil and microorganisms. These valuable findings can render significant ecological benefits for establishing healthy and stable urban forests.

Table 1 The description of forest stands.

Sample plot name	Plot size/(m × m)	Stand age/year	Latitude and longitude	
ZL	8 × 10	5	119°28′55″E, 32°26′26″N	
SSL	8 × 8	5	119°28′57″E, 32°26′24″N	
SZY	10 × 5	10	119°28′56″E, 32°26′23″N	
ZKQ	8 × 8	8	119°28′55″E, 32°26′22″N	
SBL	10 × 8	8	119°29′4″E, 32°26′10″N	
ZKG	8 × 8	5	119°29′4″E, 32°25′25″N	

Materials and Methods

Study sites and experimental design

Located at the Zhuyu Bay Scenic Area, Guangling District, Yangzhou, Jiangsu Province, the main site of the National Positioning Observation and Research Station of Urban Ecosystem was selected for this study. The climate in this area transits from subtropical humid monsoon type to temperate monsoon type, with an average annual temperature of 15.8 ∘C and an average annual precipitation of 864 mm. This region has an average altitude of 2 m, covered with zonal secondary forest vegetation. The well-grown forest stands represent the plain water network area in the lower reaches of the Yangtze River. The forest resources here are abundant, and the main types of vegetation include Cedrus deodara, Metasequoia glyptostroboides, and Phyllostachys pubescens.

Sample collection

We selected six stands for research, including Phyllostachys edulis (ZL), Metasequoia glyptostroboides (SSL), Cornus officinalis (SZY), Populus, Camellia oleifera and Clerodendrum trichotomum (ZKG), Cedrus deodara and Cupressus funebris (SBL), Yulania denudate, Eriobotrya japonica, Osmanthus (ZKQ). The bulk soil (BS) not in the vicinity of any roots near the selected plant was collected for investigation. Two plots of the same area were selected for each stand. The plots are as follows: ZL (8 m ×10 m), SSL (8 m ×8 m), SZY (10 m ×5 m), ZKG (8 m ×8 m), SBL (10 m ×8 m), ZKQ (8 m ×8 m). Five sampling points were set at the four corners and centers of each plot. After removing the superficial humus layer, a soil auger was employed to collect the soil layer at a depth of 0–20 cm. After picking out litter, fine roots, small stones, and other impurities, the sample was mixed thoroughly to minimize the impact of heterogeneity. Each sample was separately placed in a sterile plastic bag, sealed, and transported to the laboratory on ice. For microbial sequencing, the samples were stored in a refrigerator at −80 ∘C. The remaining part of the sample was used for the determination of soil physical and chemical properties. After air-drying, BS samples were ground and transmitted through a 1.0 mm sieve. Then, they were stored in a cool and dry place for soil physical and chemical property determination.

Soil physical and chemical properties

According to the previous study on Dendrobium (Yuan et al., 2020), the properties of air-dried BS samples were measured, including pH, total nitrogen (TN), total phosphorus (TP), available phosphorus (AP), and available potassium (AK). All chemical analyses were repeated three times.

DNA extraction, amplification and sequence analysis

Microbial DNA from 60 samples (from the six forest stands with ten replicates of each plot) was extracted using the CTAB/SDS method (Niemi et al., 2001). DNA concentration and purity were identified on 1% agarose gel. Depending on the concentration, DNA was diluted to 1 ng/ μL with sterile water. The primer sets 515F (5′-GTGCCAGCMGCCGCGG-3′) and 806R (5′-GGACTACHVGGGTWTCTAAT-3′) were used to generate bacterial libraries with a unique 6-nt barcode at 5′ of the forward primer to amplify the V4 region of the 16S rRNA gene for each sample. Similar to the above process, the ITS1 region of the fungus was amplified using ITS1-1F-F (CTTGGTCATTTAGGAAGTAA) and ITS1-1F-R (GCTGCGTTCTTCATCGATGC) (Caporaso et al., 2012) to generate sequencing libraries. Index codes were added with the TruSeq®DNA PCR-Free Sample Preparation Kit (Illumina, San Diego, CA, USA), complying with the manufacturer’s recommendations. Library quality was evaluated on a Qubit@ 2.0 fluorometer (Thermo Scientific, Waltham, MA, USA) and an Agilent Bioanalyzer 2100 system. After purification and quantification, the libraries were sequenced on an Illumina NovaSeq 6000 platform according to standard protocols. To treat all raw data from the 16S V4 bacterial region and the fungal ITS1 region (V1.9.1, http://qiime.org/scripts/split libraries fastq.html), QIIME and FLASH were adopted for quality-controlled process and paired reads, respectively (V1.2.7, http://ccb.jhu.edu/software/FLASH/) (Bolyen et al., 2019; Magoč & Salzberg, 2011). For bacteria and fungi, Silva sequences were matched with the UCHIME algorithm and Unite database for annotation (ITS: https://unite.ut.ee/) (UCHIME, http://www.drive5.com/usearch/manual/uchime_algo.html) (Koljalg et al., 2013; Quast et al., 2012).

Statistical analysis

The analysis of alpha diversity adopted four indices, including ACE, Chao1, Shannon, and Pielou evenness (Chao & Yang, 1993; Chao, 1984; Shannon, 1948). These indices were calculated using QIIME (Version 1.9.1) and visualized with R package ggplot2 (Version 2.15.3). One-way ANOVA was used to determine significant differences with a P-value of 0.05. If significant differences were detected, the outlier data were identified using Duncan’s post hoc test. The differences in species complexity among samples were derived from the beta diversity analysis. Moreover, the beta diversity on Bray-Curtis was obtained using QIIME software (version 1.9.1). The R language labdsv package was employed to calculate the indicators of each species in each treatment in the comparison group (Roberts & Roberts, 2016). Cross-validation was used for statistical testing to obtain the P value, which is displayed in a bubble chart.

Results

Alpha and beta diversity of urban forest stands microbial communities

Alpha diversity is an important index reflecting the number and relative abundance of species in a community. Among the four indices of alpha diversity, Chao1 and ACE are classified as richness indices; Shannon comprehensively considers the number of species and the evenness of the community; Pielou’s evenness is used the most commonly. They were calculated in this study to compare the diversity and evenness of BS microorganisms in diverse forest stands. The results exhibit that the ACE and Chao1 of the six forest stands have the same trend, with SZY being the highest and SBL being the lowest, indicating that SZY has the highest abundance of BS bacteria (Fig. 1A–Fig. 1B). In terms of Pielou’s evenness, ZL has the highest value, and SBL has the lowest, representing that the evenness of BS bacteria is the highest in ZL and the lowest in SBL (Fig. 1C). The value of Shannon in ZL is the highest, implying that ZL has the highest BS bacterial diversity (Fig. 1D). The One-way ANOVA analysis shows significant differences in the richness of BS bacteria among the six forest stands, while the differences in evenness and diversity are not significant. The alpha diversity analysis of BS fungi was performed on the six forest stands. The results show that the ACE and Chao1 of ZKQ are the highest, indicating that ZKQ has the highest abundance in BS fungi (Fig. 1E–Fig. 1F). The Pielou’s evenness and Shannon of the six forest stands present the same trend. Meanwhile, these values of ZKQ are the highest, indicating that ZKQ has the highest evenness and diversity of BS fungi (Fig. 1G–Fig. 1H). One-way ANOVA among the six forest stands discloses significant differences in the diversity and evenness of BS fungi.

Figure 1 α-diversity estimates of samples.

(A) Bacterial ACE; (B) Bacterial Chao1; (C) Bacterial Pielou’s eveness; (D) Bacterial Shannon; (E) Fungal ACE; (F) Fungal Chao1; (G) Fungal Pielou’s eveness; (H) Fungal Shannon.

In the PCA figure of bacteria, PC1 and PC2 account for 23.33% and 17.18% of the total variance, respectively (Fig. 2A). Meanwhile, ZL, SZY, and ZKG are far from the other samples, demonstrating that their bacterial communities are more different. In the PCA of fungi, the ratios of PC1 and PC2 are 21.80% and 16.90%, respectively (Fig. 2B). The differences among the bacterial communities of the six forest stands are greater than those among the fungal communities.

Figure 2 Beta-diversity in urban forest stands.

(A) Principal component analysis (PCA) of bacterial communities; (B) PCA of fungal communities.

Microbial community composition in urban forest stands

According to the OTU annotation results, the BS bacteria of the six forest stands contain 45 phyla, 138 classes, 303 orders, 447 families, and 776 genera (Fig. 3). Among them, Proteobacteria, Acidobacteria, Actinobacteria, Chloroflexi, Planctomycetes, Bacteroidetes, Verrucomicrobia, Firmicutes, Gemmatimonadet, and Patescibacteria are the dominant bacterial phyla with a relative abundance greater than 1%. The results manifest that the differences among the six forest species are increasingly evident below the phylum level. At the phylum level, Proteobacteria has the highest relative abundance in SZY. Proteobacteria and Acidobacteria in SSL, ZKG, SBL, and ZKQ have similar relative abundance. At each level, although the top ten species with higher relative abundance among the six BS bacteria are the same, their relative abundances are still somewhat different. In addition, BS fungi include 22 phyla, 54 classes, 144 orders, 290 families, and 513 genera (Fig. 4). Among them, the dominant bacterial phyla with a relative abundance greater than 1% are Ascomycota, Basidiomycota, Mortierellomycota, Anthophyta, Mucoromycota, Chlorophyta, Ciliophora, Glomeromycota, Nematoda, and Rozellomycota. The results verify that, at the phylum level, Ascomycota dominates the BS fungi of the six forest stands. Its relative abundance is the largest in ZL and ZKG and exceeds 60% in ZL and ZKG. At all levels, the relative abundance of the top-ranked bacterial group in the six forest stands exceeds 40%.

Figure 3 Top 10 relative abundances of bacterial communities in urban forest stands of bulk soil as:

(A) phylum, (B) class, (C) order, (D) family and (E) genus.

Figure 4 Top 10 relative abundances of fungal communities in urban forest stands of bulk soil as: (A) phylum, (B) class, (C) order, (D) family and (E) genus.

Differences in the abundance of OTUs in different forest stands of bulk soil microbial communities

To detect abundance differences in bacterial communities, DESeq2 was used to perform differential analysis on pairwise comparisons of BS bacteria in the six forest stands (Fig. 5). The results show that at the phylum and class levels, only SZY has unique bacterial groups. At the order level, only ZKG does not have unique bacterial groups. There are no common bacterial groups at the family level. ZL, SSL, SZY, ZKQ, and SBL have 16, 5, 22, 3, and 4 unique bacterial groups, respectively. At the genus level, common and unique bacterial groups bacterial groups both exist in the six forest stands. Additionally, differential analysis was performed on fungi. There are fifteen bacterial groups at the phylum level, of which 13 are common bacterial groups, and the remaining two are unique bacterial groups in ZKQ and SZY. At the class and order levels, the six forest stands have both common and unique bacterial groups. At the family and genus levels, there are no common bacterial groups, and the six forest stands all have unique bacterial groups.

Figure 5 Petal diagrams of bulk soil microorganisms in different forest stands at different levels.

(A) phylum, (B) class, (C) order, (D) family and (E) genus.

Identification of biomarker taxa in the bulk soil of different urban forest stands

Biomarkers refer to species or communities within a certain area that can indicate the growth environment or certain environmental conditions. Indicator analysis is generally used to screen indicator species for each group. The bacteria results represent that among the top ten biomarkers at the phylum level, there is no significant difference in the relative abundance of different BS indicator species in ZL, SSL, and ZKQ (Fig. 6). The relative abundance of various BS indicator species in SZY, ZKG, and SBL is not significant. The difference in abundance is significant. Among them, the relative abundance of Firmicutes in ZKG is the lowest. Moreover, below the class level, the relative abundance of different indicator species in each forest stand is significantly different. The indicator analysis of fungi shows that their relative abundance and bacterial community differ significantly (Fig. 7). The results indicate that the relative abundance of indicator species of BS fungi is generally low at all levels. At the phylum level, relatively abundant indicator species include Nematoda and Anthophyta in SZY, Mucoromycota in SBL, and Glomeromycota in ZKQ. Furthermore, SZY and SBL have indicator species with relatively high abundance at all levels. The relative abundance of Nematoda is only high in SZY and low in the other five forest stands

Figure 6 Biomarker bubble diagram of bacterial at different levels in different forest stands.

The size of the bubble indicates the relative abundance of the indicator species. (A) Phylum, (B) class, (C) order, (D) family and (E) genus.

Figure 7 Biomarker bubble diagram of fungal at different levels in different forest stands.

The size of the bubble indicates the relative abundance of the indicator species. (A) Phylum, (B) class, (C) order, (D) family and (E) genus.

Figure 8 Three-dimensional ranking of soil bacteria (A) and fungi (B) at phylum level and soil physical and chemical properties in different forest stands.

Table 2 Physiochemical properties of Soil from different forest stands.

Sample plot name	pH	SOM (g/kg)	TN (g/kg)	TP (g/kg)	AK (g/kg)	AP (mg/kg)	
ZL	8.26 ± 0.37b	8.63 ± 2.28c	0.75 ± 0.07b	0.94 ± 0.07a	0.35 ± 0.031b	37.64 ± 2.28a	
SSL	8.11 ± 0.08b	13.23 ± 1.30b	0.81 ± 0.02b	0.73 ± 0.17ab	0.22 ± 0.03c	23.76 ± 1.34b	
SZY	8.91 ± 0.29a	4.15 ± 1.27d	0.39 ± 0.03c	0.79 ± 0.06ab	0.22 ± 0.01c	15.99 ± 0.56d	
ZKG	7.14 ± 0.07c	5.66 ± 0.51d	0.72 ± 0.22b	0.77 ± 0.13ab	0.36 ± 0.02b	19.97 ± 2.36c	
SBL	6.26 ± 0.30d	4.42 ± 1.00d	0.60 ± 0.20bc	0.65 ± 0.12b	0.43 ± 0.06a	24.82 ± 1.10b	
ZKQ	6.68 ± 0.16d	17.52 ± 1.59a	1.10 ± 0.17a	0.90 ± 0.14a	0.30 ± 0.03b	39.56 ± 3.13a	

Relationships between bulk soil microbiomes and soil factors

In order to analyze the relationship between samples, microbial communities, and environmental factors, the redundancy analysis (RDA) was performed on BS samples (Fig. 8). The results exhibit that SOM, AP, and pH change greatly in BS microbial communities. Among them, SOM is positively correlated with the changes in the BS bacterial community and fungal community of SSL. The changes in the BS bacterial communities and fungal communities of ZL, SBL, and ZKQ are affected by AP. Meanwhile, soil pH influences the BS of SZY and ZKG. Important factors change in bacterial and fungal communities. In addition, Anthophyta and Basidiomycota impact changes in the BS fungal communities of SSL and SZY. Environmental factors have evident effects on BS bacterial communities and fungal communities. The determination of soil physical and chemical properties shows that the values of SOM, TN, AK, and AP in SZY are relatively low (Table 2). There is no significant difference in the values of soil SOM of SZY, ZKG, and SBL, all lower than those of ZL, SSL, and ZKQ, and the latter three have significant differences. The values of TP of the six forest stands have no significant difference. The difference in TN between ZKQ and SZY is significant, while there is no significant difference in TN between ZL, SSL, ZKG, and SBL (Table 2). Moreover, the differences in pH, AK, and AP among the six forest stands are significant, indicating that factors affecting the soil environments of the six forest stands are different.

Discussion

Vegetation diversity impacts urban forest BS microbial communities. The alpha diversity analysis of BS bacterial and fungal communities in the six forest stands reveals that among the bacterial communities, ZL has higher ACE and Chao1 indices and the highest Pielou’s evenness and Shannon indices, implying that ZL has the highest bacterial evenness and diversity and relatively high bacterial richness. Relevant research shows massive litter in bamboo forests, decomposed by soil microorganisms to obtain nutrients for their growth and reproduction. This may be one of the reasons for the high diversity of bamboo forests. Among the fungal communities, ZKQ has the highest richness, evenness, and fungal diversity. Ascomycota is found to be dominated by saprotrophic bacteria, which can decompose various refractory substances. Meanwhile, Basidiomycota can degrade lignin and other refractory substances. ZKQ has richer vegetation types, with more surface litter and lignin, providing a favorable environment for the large-scale reproduction of Ascomycota and Basidiomycota.

This study found a total of 45 phyla, 138 classes, 303 orders, 447 families, and 776 genera in bacterial communities. The fungal community consists of 22 phyla, 54 classes, 144 orders, 290 families, and 513 genera. A total of 33 phyla, 79 classes, 156 orders, 290 families, 474 genera of soil bacteria and eight phyla, 25 classes, 84 orders, 185 families, and 360 genera of soil fungi were detected in the Harbin urban forest, which belongs to the mid-temperate zone (Ao et al., 2022). Multiple factors contribute to the differences in microbial community structures between the two places. Temperature and moisture content have the most important effects on the growth and activity of microorganisms in the soil. It has been reported that when the temperature is high, plant metabolism is strong and will release secretions into the soil, stimulating the reproduction of soil bacteria. Moisture mainly affects the diffusion of microbial respiratory substrates. In particular, the movement of bacteria in the soil environment and the process of obtaining nutrients rely on the flow of water films in the soil. Therefore, moisture has a greater impact on fungi.

The results of this study verify that the dominant phyla of the BS bacterial communities in the six forest stands are Proteobacteria, Acidobacteria, Actinobacteria, Chloroflexi, Planctomycetes, Bacteroidetes, Verrucomicrobia, Firmicutes, Gemmatimonadet, and Patescibacteria, reflecting the richness and diversity of the BS bacterial communities in the six forest stands. Among them, the primary ones are Proteobacteria, Acidobacteria, and Actinobacteria. It is consistent with the results of other studies (Ao et al., 2022). Proteobacteria has the highest relative abundance in the soil of the six forest stands and the bacteria that chiefly decompose and transform litter (Barnard, Osborne & Firestone, 2013; Harichová et al., 2012; Peiffer et al., 2013). However, the relative abundance of dominant bacterial phyla and bacteria in the BS fungal communities of the six forest stands are quite different. Ascomycota has the highest relative abundance, which agrees with existing research results. The relative abundance of Basidiomycota is high in SSL and SBL. Basidiomycota is the main decomposer of soil fungi and can decompose lignocellulose. Relevant studies have discovered that Ascomycota and Basidiomycota are the dominant phyla of urban soil fungi in Dongguan (Tan et al., 2019).

Moreover, this study revealed that soil physical and chemical properties had a significant impact on soil microorganisms (Huang et al., 2019; Wang, Wu & Kumari, 2018). pH, SOM, and AP play a crucial role in affecting soil microbial communities. Soil pH influences soil metabolic function, microbial community structure, and the availability of soil organic matter and nutrients. The results manifest that under the influence of environmental factors, the bacterial diversity of the six forest stands is higher than the fungal diversity. It aligns with related studies (Shen et al., 2013). The RDA analysis shows that SOM is positively correlated with the soil bacterial community in SSL, and the highest relative abundance in SSL is Proteobacteria, which may be consistent with the positive correlation between Proteobacteria and SOM (Navarrete & Tsutsuki, 2008). AK, TN, and TP also have remarkable effects on the compositions of the BS microbial communities (Van Geel et al., 2016; Yoshimura et al., 2013).

Conclusion

This study explored differences in BS microbial communities in diverse forest stands. Among the six forest stands, ZL has the highest bacterial diversity, while ZKQ has higher fungal diversity. Meanwhile, the relationship between environmental factors and soil microbial communities was explored. SOM, AP, and pH are key environmental factors that affect changes in BS microbial communities. These findings will contribute to future in-depth research on soil microbial diversity in urban forests and can help policymakers formulate sustainable and ecologically beneficial urban landscapes. This study can provide valuable information on ecosystem services, spatial planning and land use, urban ecological planning, and biodiversity conservation and render a scientific basis for decision-making.

Supplemental Information

Supplemental Information 1 Raw data for Table 2

Additional Information and Declarations

Competing Interests

Author Contributions

Data Availability

The authors declare there are no competing interests.

Xin Wan conceived and designed the experiments, performed the experiments, analyzed the data, prepared figures and/or tables, authored or reviewed drafts of the article, and approved the final draft.

Runyang Zhou performed the experiments, analyzed the data, prepared figures and/or tables, authored or reviewed drafts of the article, and approved the final draft.

Yingdan Yuan performed the experiments, analyzed the data, prepared figures and/or tables, authored or reviewed drafts of the article, and approved the final draft.

Wei Xing conceived and designed the experiments, prepared figures and/or tables, authored or reviewed drafts of the article, and approved the final draft.

Sian Liu conceived and designed the experiments, prepared figures and/or tables, authored or reviewed drafts of the article, and approved the final draft.

The following information was supplied regarding data availability:

The data is available at NCBI SRA: PRJNA1029967 (bacterial) and PRJNA1030501 (fungal).

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
