# Peer review of "Microbiota associated with urban forests"

_PeerJ, doi:10.7717/peerj.16987_

## Round 0.1 · original submission · Major Revisions

The authors are required to revise the manuscript as per the suggestions of the reviewers.

Reviewer 1 ·

Basic reporting

no comment

Experimental design

1- It is better to describe the tree species within the following plots: Mixed broad-leaved shrub forest
Mixed pine and cypress forest, Mixed broad-leaved tree forest, since the soil's biological properties are highly affected by the residues of the leaves that fall from tree species to the ground.

2-What reasons do the authors provide for utilizing four indices of species diversity (ACE, Chao1, Shannon, and Pielou evenness)? Could they explain the key distinctions between these indices? And also write the components of each index.

Validity of the findings

The MN missing the conclusion section and the authors have to add this section

Additional comments

I have read the manuscript “Microbiota associated with the soil of forest stands in urban forests” and I found the authors addressed important questions What commonalities and variances exist in the composition of microbial communities in the bulk soil across six distinct forest stands? (2) How are soil factors related to the composition of microbial communities in the bulk soil?

This study investigated the soil microbial communities in various urban forest locations to understand the interaction between soil and microorganisms. The findings are valuable for establishing healthy and stable urban forests, offering significant ecological benefits.

The MN is well-written and states sound objectives. The methodology and the statistical analysis are accurate and well-employed for the data set. However, I have some comments which need to be assessed in a revised version of the manuscript. I have also provided several comments within the main text of the MN that should be taken into account.

Comments

Title: Is good, concise, clear, and informative. It captures the attention of the readers. However, I suggest removing of “forest stands in “

Abstract: Present clearly the experimental methods and results. However, the conclusion paragraph needs to effectively address the meaning of the findings.

Keywords: almost all the keywords were employed in the title, is better to use another keywords this will makes the paper more searchable.

Introduction:
Overall, the introduction is well-developed situates the readers in the context of the study, and ends by showing the reader clearly the questions of the study.


Material and Methods
This part is present clearly the experimental setting and taking measurements. However,
1- The analysis should involve an increased number of plots within each forest stand, rather than relying on a single plot.
2- It is better to describe the species within the following plots: Mixed broad-leaved shrub forest
Mixed pine and cypress forest, Mixed broad-leaved tree forest, since the soil's biological properties are highly affected by the residues of the leaves that fall from tree species to the ground.
3-What reasons do the authors provide for utilizing four indices of species diversity (ACE, Chao1, Shannon, and Pielou evenness)? Could they explain the key distinctions between these indices?



The results
Generally, the results were presented in an organized and clear way and described well what is shown in the tables and figures. The results are accurate, reliable, and trustworthy. The research findings are valid and credible within the context of the study. The authors used various methods and techniques to establish the validity of their findings, including rigorous experimental design, data analysis, and thorough interpretation of results.

Discussion

This section needs to be improved since there is no sufficient discussion for the main obtained findings but more re-displaying the results.


Conclusions
The Manuscript missing this section.

Tables:
Table 1: Sample plot name, here the authors need to refer to the used abbreviations below Table 1
Forest age/year is Stand age/year
Table 2: the title of the first column is missing ( Sample plot name)

Figure: all the captions should be below the figures.

Annotated reviews are not available for download in order to protect the identity of reviewers who chose to remain anonymous.
Cite this review as

Reviewer 2 ·

Basic reporting

The manuscript “Microbiota associated with the soil of forest stands in urban forests” deals with the research on urban forest soil microbial communities is valuable for understanding the ecological dynamics in urban environments.
Comments
• The study deals with the high-throughput sequencing technology is an appropriate and advanced method for studying soil microorganisms, providing a comprehensive understanding of microbial communities.
• LN 18-21; Please rewrite the line.
• Could the study benefit from a more detailed analysis of the specific species within the identified bacterial and fungal communities? Please provide in the introduction section
• The identification of Proteobacteria as the most abundant bacterial phylum is consistent with existing literature, validating the reliability of the results.
• LN 41: Please check the reference and its styling.
• Were there any temporal aspects considered in the study, such as seasonal variations, which might influence microbial community dynamics?
• Ln 66-71: Please rewrite the content.
• The objectives of the study need to be revisited and rechecked.
• How might the results of this study inform land-use planning and urban development strategies to better support and protect microbial diversity in urban forests?
• Details about the sampling methodology, such as the number and location of sampling points within each forest stand, should be provided in the material and method section.
• The statistical methods used for analyzing the microbial community data should be clearly outlined. It would be helpful to know how the alpha diversity and community differences were assessed and if any corrections for multiple comparisons were applied.
• The exploration of urban forest stands has practical implications for policymakers and urban planners aiming to create sustainable and ecologically beneficial urban landscapes.
• Ln 245-249; Please rewrite the line.
• Could the study benefit from a more in-depth investigation into the functional roles of specific microbial taxa within the urban forest ecosystems?
• In what ways do the environmental factors (soil pH, organic matter, available phosphorus) interact and potentially synergize or counteract each other in shaping microbial communities? Please provide details in the discussion section.
• Could the study benefit from a broader geographical scope to explore how urban forest microbial communities vary across different regions and climates of other countries of agroecological zones?
• The conclusion section is missing from the text.
• Consider suggesting potential future research directions based on the findings in the conclusion section.

Experimental design

Please see 1. Basic reporting

Validity of the findings

Please see 1. Basic reporting

Additional comments

Please see 1. Basic reporting

Cite this review as

---

## Round 0.2 · accepted · Accept

The manuscript is improved and acceptable in its current state.

Reviewer 1 ·

Basic reporting

The revised manuscript is now clear and unambiguous, thanks to enhancements in article structure, figures, tables, and the provided raw data. There has been a notable improvement in the use of English throughout the document. The manuscript is currently more self-contained, presenting relevant results aligned with the hypotheses. I believe the manuscript aligns well with the journal requirements and is ready for publication.

Experimental design

The authors have adequately addressed the comments made by me for this part in the
revised version of the manuscript. Therefore, I have no further comments.

Validity of the findings

All underlying data have been supplied, demonstrating robustness, statistical soundness, and effective control.
Conclusions are well-formulated, directly tied to the original research question, and appropriately restricted to supporting results.

Additional comments

The authors have adequately addressed the comments made by me in the revised version of the manuscript. Therefore, I have no further comments.

Cite this review as

Reviewer 2 ·

Basic reporting

The author made significant changes in the manuscript.

Experimental design

see section 1

Validity of the findings

see section 1

Additional comments

see section 1

Cite this review as